# Forecasting for Chaotic Time Series Based on GRP-lstmGAN Model: Application to Temperature Series of Rotary Kiln

**DOI:** 10.3390/e25010052

**Published:** 2022-12-27

**Authors:** Wenyu Hu, Zhizhong Mao

**Affiliations:** School of Information and Sciences, Northeastern University, Shenyang 110819, China

**Keywords:** chaotic time series, global recurrence plot, generative adversarial network, long short-term memory

## Abstract

Rotary kiln temperature forecasting plays a significant part of the automatic control of the sintering process. However, accurate forecasts are difficult owing to the complex nonlinear characteristics of rotary kiln temperature time series. With the development of chaos theory, the prediction accuracy is improved by analyzing the essential characteristics of time series. However, the existing prediction methods of chaotic time series cannot fully consider the local and global characteristics of time series at the same time. Therefore, in this study, the global recurrence plot (GRP)-based generative adversarial network (GAN) and the long short-term memory (LSTM) combination method, named GRP-lstmGAN, are proposed, which can effectively display important information about time scales. First, the data is subjected to a series of pre-processing operations, including data smoothing. Then, transforming one-dimensional time series into two-dimensional images by GRP makes full use of the global and local information of time series. Finally, the combination of LSTM and improves GAN models for temperature time series prediction. The experimental results show that our model is better than comparison models.

## 1. Introduction

Rotary kiln sintering temperature (ST) is one of the basic parameters in the system, which is a key and global controlled parameter to the sintering technology and determines the quality of the production and energy consumption. Therefore, research on ST is highly valuable for industrial applications.

ST influenced by many uncertain factors, such as the composition of the raw material and the structure of the rotary kiln. However, it also has complex nonlinear characteristics caused by physical and chemical changes. Therefore, the high-precision prediction method of the ST is a hot topic in academic circles. Early investigations used the conservations of mass and energy or the physicochemical equation to simulate ST [1]. With the development and application of soft sensor methods in the process industries, numerous cutting-edge techniques have been used to anticipate and measure ST, such as data-driven methods  [2] and sintering image analysis [3,4].

Nevertheless, these techniques disregard the inherent nonlinear pattern in ST. The concepts of chaos theory and nonlinear dynamic analysis can not only enhance the understanding of the fundamental characteristic of time series but also help to extend the predictive capability. Lv made the first mention of the sintering system of rotary kiln is a fifth-order chaotic system [5,6]. The traditional prediction method of chaotic time series is based on the assumption that real systems are recursive, that is, the future state is the same or similar to a previous state. For example, in [7], two multivariate time series phase space reconstruction methods are established; the authors implement wind power forecasting using the least-square support vector machine algorithm. In [8], a weighted largest Lyapunov exponent method was presented to enhance prediction accuracy.

As opposed to the standard chaotic time series forecasting process, which involves fitting a model to the historical data and simulating future data utilizing fitted model, selecting the most appropriate forecasting model or averaging several models has been a popular alternative approach. For example, in [9], researchers proposed a hybrid model of a stacked autoencoder and modified particle swarm optimization for multivariate chaotic time series forecasting. In  [10], the author developed the recurrent predictor neural network for predicting chaotic time series, which is more accurate than the universal learning network. In [11], the authors presented a gate recurrent unit-based deep recurrent neural network to forecast the time series of Lorenz, Rabinovich–Fabrikant, and the Rössler chaotic system, which presented better performance than the long short-term memory (LSTM) model.

Reviewing the literature on chaotic time series forecasting, we find that (i) most of the present approaches depend on the manual choice of an appropriate set of features, which determines the prediction accuracy that relies on the feature-extraction method and characteristics of data. More significantly, (ii) the current literature focuses on the local features of the time series, leaving global characteristics under-emphasized. For the ST with a large time lag, the global dynamics of the time series contain important information such as irregular physical and chemical changes changes of sintered materials. Inspired by the recent work of [12,13,14], this paper aims to explore chaotic time series forecasting based on the idea of GRP, from which the global and local features of the time series can be automatically extracted using computer vision algorithms. The main contributions of this paper are as follows:This paper proposes to use GRP to transform the series into a two-dimensional image, which gives a priori knowledge about similarity and predictability by explaining the global and local information of the time series.Compared with WGAN-gp, our GAN model employs one generator and two discriminators, which enhances the convergence speed of the GAN and generates a more realistic GRP.A new chaotic time series forecasting model is put forward, which combines GAN and LSTM to automatically extract the features of the time series by the convolutional neural network (CNN). It achieved the highest prediction accuracy compared to the other models.Most existing models do not consider the time delay between the model input and output variables; however, when using multiple variables for prediction, it is important to consider the correlation between time series. In this paper, we find that choosing the appropriate time delay can improve the prediction accuracy through experiments.Images cannot be used as an input to LSTM. Therefore, a suitable image feature-extraction method is needed to not only reduce the input vector dimension but also to reserve as much information as possible. This paper analyzes the impact of different feature-extraction methods on the prediction accuracy.

The structure of the rest of this paper is as follows: Section 2 formulates the temperature of rotary kiln predicting problem and introduces the background of our model. Section 3 describes our forecasting model. Section 4 describes various evaluation experiments and gives a qualitative discussion of the experimental results. Section 5 is conclusion. List of abbreviationsacronyms used in this article.

## 2. Preliminary Knowledge

### 2.1. Formulation of Temperature of Rotary Kiln Predicting Problem

The goal of predicting the problem of the ST is to use previous data to predict the future temperature. $t_{n}$ denotes the temperature data at moment *n*. Then, the control variable at moment *n* is represented by $x_{n}$. We document a group of consecutive observed data as $t_{0},t_{1},t_{2},\cdot \cdot \cdot$. The problem is to forecast the temperature value at a certain point in the future with *j* previous temperature data and control variables by continuous observation:(1)$$\begin{array}{c} {\hat{t}}_{n+\tau }=\underset{t_{n+\tau }}{\arg\max}p\left(t_{n+\tau }|f\left({\hat{t}}_{n-j+1},{\hat{t}}_{n-j+2},..,{\hat{t}}_{n}\right),{\hat{x}}_{n-j+1},{\hat{x}}_{n-j+2}\right). \end{array}$$
where p() denotes predictive model and f() stands for pre-processing of previous temperature data. Before describing our approach in detail, a brief introduction to the basics is needed.

### 2.2. Phase Space Reconstruction

In our model, the PSR is the fist step to the pre-processing of a rotary kiln temperature time series, which can reflect the richer information of the chaotic time series. For the one-dimensional scalar time series x(i):1≤i≤N with infinite length, no noise, and a *d* dimensional chaotic attractor, an *m* dimensional embedded phase space can be found in the sense of topology invariance. Takens theoretically proved that the delay coordinate vector space is the differential homeomorphism of the attractor of the original dynamical system in the Euclidean space $R^{m}$ if the dimension *m* of the delay coordinate vector is greater than twice the attraction dimension *d* of the original dynamical system [15].

The delay-coordinate reconstruction of the high-dimension dynamic system from time series by two important parameters, time delay τ and embedding dimension *m*. The State *y* on phase space trajectory.
(2)$$y_{1},y_{2},..,y_{l}$$
where l=N−(m−1)τ and
(3)$$y_{j}=\left(x_{j},x_{j+\tau },..,x_{j+(m-1)\tau }\right).$$

There are two dominant views on finding the embedding dimension *m* and the delay time τ. The first view is that *d* and *m* are mutually uncorrelated and that the appropriate embedding dimension should be found after finding the delay time. Typical methods for finding time delays are the autocorrelation method [16] and the mutual information method [17]. The method for finding the optimal embedding dimension is the false nearest neighbour (FNN) [18].

### 2.3. Recurrence Plot

Based on the PSR, RP can convert one-dimensional chaotic time series into two-dimensional matrices to extract more enrichment information using convolutional neural networks. The recurrence of the STates is a fundamental property for deterministic dynamic systems with a certain degree of nonlinearity. Such a new graphical tool was first proposed for time series analysis by [19]. The recurrence plot was used to visualize the recurring patterns, which display important interpretable information about time scales. It interprets the internal structure of time series, gives a priori knowledge of similarity and prediction, and is an important method for analyzing the recurrence, chaos, and non-stationarity of time series. With the help of CNN, we can extract local and global information of chaotic time series from RP. The RP can be expressed by the following equation:(4)$$R_{ij}=\Theta (\varepsilon -∥x_{i}-x_{j}∥).$$

For time series x(i):1≤i≤N. After the relevant theoretical calculations to determine the appropriate embedding dimension and delay time and then reconstruct the time series, the reconstructed dynamical system is:(5)$$u_{i}=\left[x_{i},x_{i+\tau },..,u_{i+(m-1)\tau }\right].$$
where i=1,2,..,N−(m−1)τ. Calculate the distance between $x_{i}$ and $x_{j}$ in the reconstructed phase space.
(6)$$S_{ij}=∥u_{i}-u_{j}∥.$$
where i=1,2,..,N−(m−1)τ, j=1,2,..,N−(m−1)τ. ∥·∥ stand for the norm. Calculate recurrence values.
(7)$$R\left(i,j\right)=\Theta \left(\varepsilon_{i}-S_{ij}\right).$$

Select a suitable threshold. Θ(·) stand for the Heaviside function:(8)$$f\left(x\right)=\left\{\begin{array}{cc} 1, & x⩾0\\ 0, & x<0 \end{array}\right.$$

However, in practical applications, it is difficult to find a suitable threshold; the unthreshold recursive graph is intrinsically a direct expression of the distance matrix, which measures the similarity between two-phase space vectors.
(9)$$GRP_{ij}=∥u_{i}-u_{j}∥.$$

### 2.4. Generative Adversarial Networks

In our method, GAN is used to generate the GRP ($rp\in R^{N\times N}$) of the temperature data from a set of random noise. We use $z∼p_{z}$, which follows a uniform distribution with a maximum value of 1 and a minimum value of −1, as the hidden variable. Then generator is defined as $G(z;\theta_{g})$. It maps the random noise to data space ($rp\in R^{N\times N}$), where rp=G(z). The discriminator is described as $D(x;\theta_{d})$. The input of *D* is a real data *x* sample from $P_{data}$, a fake sample generated by *G*, or a fake sample generated by *G*. The output of *D* represents the probability of its considering the picture to be real or unreal, expressed as a scalar from 0 to 1. The basic structure of GANs is shown in Figure 1. GAN is essentially a transformation or mapping from hidden space to sample space.

## 3. GRP-lstmGAN Model for Temperature Predict

We now present our GRP-LBP-lstmGAN model for ST forecast (see Figure 2). Firstly, the GRP of the ST is used for prediction rather than the temperature itself or the state after PSR. Since ST is a complex and chaotic time series, it is difficult to interpret and acquire complex nonlinear information from the data itself. Using PSR, we can reconstruct an equivalent system in state space by using the observed chaotic time series. The time delay and dimension embedding can be used to reconstruct the influence of other variables on the system, and GRP can show significant and easily explained local and global information about the time scale [19,20,21]. In addition to when a one-dimensional time series is transformed into an image, the noise is considered to be the illumination of the image by light, thus eliminating the effect of noise on the signal [22]. Second, We use the local binary pattern (LBP) to preprocess the GRP images before sending them into the network. Microscopic patterns such as the lines parallel to the main diagonal, the lines perpendicular to the main diagonal, the vertical (horizontal) lines, and the isolated points in the GRP reflect the characteristics of the time series, e.g., when two neighboring orbits in phase space are continuously approaching one another during a period of evolutionary time, the lines parallel to the main diagonal appear, and the length of the lines reflects the speed of the separation of the neighboring orbits. LBP can extract these microscopic patterns easily and rapidly.

Finally, the LBP of the GRP used as the input to lstmGAN, which combines the image generation capability of the GAN and the time series memory capability of LSTM. The LSTM model extracts the features of the GRP pattern moving along the main diagonal. The generator of the GAN transforms the output of LSTM into the new RP, and the discriminator of the GAN automatically extracts the features of the RP for predicting the ST. The training procedure can be separated into two steps. At first, the real GRP of the ST is use to train the GAN. Secondly, when the GAN is well trained. the LSTM and the GAN are connected together. However, the parameters of the generator of the GAN will be fixed. Because the number of parameters of the LSTM is smaller than the generator network, if we train them at once, the generators will play a dominant role. During the training process, the neural network will tend to discriminate whether the texture features of the input image are real or not rather than predicting the future temperature [14].

## 4. Data Source

The data comes from the thermocouple data of No.5 kiln of Zhong-Zhou Aluminum Plant in Jiaozuo, Henan Province. Data length 205 h, sampling frequency 60 hz. Table 1 exhibits the statistical of the data set.

Figure 3 shows an example of the datasets for the model. Firstly, every 60 pieces of temperature data are averaged and smoothed (e.g., Data[0]). Secondly, one GRP is calculated for each 157 data (e.g., GRP[0]). Finally, the input x to the neural network consists of the texture features of the GRP (LBP for GRP, e.g., $X_{train}\left[0\right]$) and the control variables (the fuel consumption of rotary kiln, e.g., $Y_{train}\left[0\right]$).

## 5. Experiments

In this section, we designed experiments to fulfill the following goals:To allow the WGAN-gp to generate a more realistic method than the original method.To compare the accuracy of our model with other prediction models for ST prediction.To improve the prediction accuracy by analyzing the time delay characteristics of the input and output.To improve the prediction accuracy by analyzing the features of different feature-extraction methods.

### 5.1. Compare of the GANs

The convergence problem of the GAN is still a pressing issue. The same model performs differently on varied datasets. The WGAN-gp trains faster and generates sharper images than the original GAN. However, the GRP is symmetrical strictly according to the main diagonal. To solve this problem, we propose to improve the WGAN-gp for RP (see Algorithm 1). The difference compared with the original one is that the loss function of the generator is divided into two parts. In the first part, the discriminator is the same. In the second part, the discriminator is to judge whether the output picture of the generator is symmetrical along the main diagonal.
(10)θG←RMSprop(∇θG1m∑i=1m−DθD(GθG(z))−myloss,θG)wheremyloss=1−iIteration∗1n∗n∗(∣GθG(z)−GθG(z)T∣)
where n∗n indicates the number of GRP pixel points and GθG(z)−GθG(z)T represents the difference between the image output by the generator and the image transposition.
**Algorithm 1** Improve WGAN-gp for RP Training Algorithm**Require:** The batch size m=32, the number of discriminator iterations per generator     iteration nD=5, the gradient penalty coefficient λ=10, the learning rate lr=0.0001**Require:** Initial generator parameters θG, initial discriminator parameters θD
**while**θG has not converged **do**
    **for** t=1,···,nD **do**
        **for** i=1,···,m **do**
           Sample real data $x∼P_{data}$, a random number.
           Sample latent variable $z∼P_{z}$.
           Sample ϵ∼U[0,1].
           $\tilde{x}\leftarrow G_{\theta_{G}}\left(z\right)$
           $\hat{x}\leftarrow \epsilon x+\left(1-\epsilon \right)\tilde{x}$
           $L^{\left(i\right)}\leftarrow D_{\theta_{D}}\left(\tilde{x}\right)-D_{\theta_{D}}\left(x\right)+\lambda (\|\nabla_{\hat{x}}D_{\theta_{D}}\left(\tilde{x}\right){{\|}_{2}-1)}^{2}$
        **end for**
        $\theta_{D}\leftarrow Adam\left(\nabla \theta_{D}\frac{1}{m}\sum_{i=1}^{m}L^{\left(i\right)},\theta_{D}\right)$
    **end for**
    Sample a batch of latent variables ${\left\{z^{\left(i\right)}\right\}}_{i=1}^{m}∼P\left(z\right).$
    $\theta_{G}\leftarrow Adam\left(\nabla_{\theta_{G}}\frac{1}{m}\sum_{i=1}^{m}-D_{\theta_{D}}\left(G_{\theta_{G}}\left(z\right)\right)-myloss,\theta_{G}\right)$
    where $myloss=1-\frac{i}{Iteration}\frac{1}{n*n}*\left(|G_{\theta_{G}}\left(z\right)-G_{\theta_{G}}{\left(z\right)}^{T}|\right)$
**end while**


To quantitatively describe whether the GRP generated by generator is symmetric or not, we define the symmetry degree:(11)$$SD=\frac{1}{n*n}*\left(|G_{\theta_{G}}\left(z\right)-G_{\theta_{G}}{\left(z\right)}^{T}|\right)$$ The smaller the SD, the better the symmetry of GRP.

According to the result shown in Figure 4, we can draw the following conclusions.

The Improved WGAN-gp for GRP has superior performance. Our proposed model can generate more symmetric GRP graphics. Firstly, the SD of our model is less than 0.1 after 100 iterations, but the original WGAN-gp requires 700 iterations. Secondly, the minimum value of SD for WGAN-gp is greater than our model. Finally, the training process of our model is more stable, without significant large fluctuations in SD.

It is not enough for GAN to generate symmetric graphs; in our task a good GAN should generate the most realistic GRP. Figure 5 shows the GRP generated by WGAN-gp and our model in different iterations. First of all, our model can generate clear images at 400 iterations; however, WGAN-gp needs 800 iterations. Moreover, in 5000 iterations, the GRP generated by our model is more realistic.

### 5.2. Accuracy of GRP-LSTMGAN Prediction

The objective in this part is to compare the accuracy of our proposed prediction model with others. In order to adequately test the superiority of our model, three algorithms are considered to compare the temperature-forecasting accuracy. These algorithms are the PSR-LSTM, the RP-LSTM, and the RP-LSTM-autoencoder, respectively. Based on the results shown in Table 2, the following conclusions can be drawn.

By comparison with other methods, the proposed model has the highest prediction accuracy, and all of the indicators are optimal.The precision of all of the methods using GRP is significantly higher than that using PSR. Further comparison of the precision of PSR-LSTM and GRP-LSTM shows that GRP provides more local and global information about the time series to improve the precision of the predictor compared to PSR (LSTM uses the same architecture, hyperparameters, and optimizer).The GRP-LSTM-Autoencoder and the proposed model use the same architecture. For comparison, our proposed model is trained using Algorithm 2, and the GAN of GRP-LSTM will not be pre-trained. The lower error shows that our proposed training method can take advantage of the GAN image generation to strengthen the model-prediction accuracy.

**Algorithm 2** GRP-LSTMGAN Training Algorithm**Require:** The batch size m=16, the learning rate lr=0.0005**Require:** Initial LSTM parameters θlstm,
Train GAN with Algorithm   1.
Connecting GAN and LSTM, freeze $\theta_{G}$ parameters.
**for** number of training iterations **do**
    Sample ${T^{\left(i\right)}}_{i=1}^{m}$ a batch from the pre-processing data.
    $T^{\left(i\right)}=t_{i-k+1},t_{i-k+2},\cdot \cdot \cdot ,t_{i}$
    $t_{i}=f\left({\hat{t}}_{n-j+1},{\hat{t}}_{n-j+2},..,{\hat{t}}_{n}\right),$
             ${\hat{x}}_{n-j+1},{\hat{x}}_{n-j+2},..,{\hat{x}}_{n}$
    $D_{\theta_{lstm}}\leftarrow \nabla_{\theta_{lstm}}\left[\frac{1}{m}\sum_{i=1}^{m}logcosh\left(D\left(G\left(f_{lstm(T^{\left(i\right)})}\right)\right),{\hat{t}}_{n+\tau }\right)\right]$
    $\theta_{lstm}\leftarrow \theta_{lstm}-lr\cdot RMSprop\left(\theta_{lstm},D_{\theta_{lstm}}\right)$
**end for**



(12)
$$\rho =\frac{\Sigma_{i}\left(x_{i}-\bar{x}\right)\left(y_{i}-\bar{y}\right)}{\sqrt{\Sigma_{i}{\left(x_{i}-\bar{x}\right)}^{2}\Sigma_{i}{\left(y_{i}-\bar{y}\right)}^{2}}}$$


By analyzing the data in Table 3, it can be concluded that our model has the highest accuracy compared to other delay times. On the one hand, the input data do not contain information about the high-delay-time temperature, which leads to a higher error in the high-delay-time prediction. On the other hand, the input data with low time delay contains redundant information, which will affect the prediction.

### 5.3. The Effect of Time Delay

When using coal consumption as control variable to predict temperature, to obtain better prediction results, the temperature time series should have an appropriate time delay, which is important for the accuracy of the degree prediction and can improve the time scale of the prediction. The temperature time series and the coal-consumption time series generally have a strong correlation as it is demonstrated that the temperature time series will be very similar to the present coal-use time series after a period of delay. The Spearman correlation coefficient was used to quantitatively measure the correlation (see Figure 6). The Spearman correlation coefficient is defined as the Pearson correlation coefficient ρ between the rank variables.

### 5.4. Comparing The Effects of Different GRP Feature-Extraction Methods on Forecast Precision

We dimensionalize the image before sending it into the LSTM. If the input vector is too large, the convergence of the model will be very slow. In this section, we compare the impact of several LBP models Table 4 on prediction accuracy.

The comparison results are shown in Table 5, and the following conclusions can be drawn. First of all, the uniform pattern is the most widely used; however, it has not obtained the best prediction accuracy. In the second place, the prediction accuracy of patterns with rotational invariance are low, probably because they cannot distinguish lines parallel to the main diagonal and horizontal (vertical) lines in the GRP. In the end, the default pattern achieves the highest accuracy, although the 256-dimensional feature vector may lead to excessive differences in the values of different features and cause errors. However, such a pattern can still extract more information from the GRP.

## 6. Conclusions

In this paper, we have proposed to use GRP to transform the 1D chaotic time series into 2D images to obtain the global and local features and construct a GRP-lstmGAN prediction model with higher prediction accuracy than the existing methods. To improve the forecast accuracy, firstly, with respect to the traditional GAN with only one discriminator, the GAN was improved by adding a non-trainable discriminator, which determined whether the GRPs generated by generator are symmetric. The experiments showed that this method improves the model convergence speed and generates more realistic GRPs. Secondly, the correlation between the input variables coal consumption and temperature series was investigated; the highest correlation with the current coal consumption was obtained when the temperature series lagged 178 units, i.e., the model achieved the highest accuracy when predicting the temperature after 178 units of time by using the coal consumption and temperature data at the current moment. Finally, directly using the GRP image as an input to LSTM, the large input dimension will prevent the model from easily converging. In this paper, the experiments have compared different feature-extraction methods; LBP-default reduced the input from 112×112 of 2D image to 256 dimensions and had the highest prediction accuracy compared with other methods.

In our model, the performance of the GAN will affect the prediction accuracy. Knowing how to avoid the pattern collapse of the GAN is the future research direction.

## Figures and Tables

**Figure 1 entropy-25-00052-f001:**
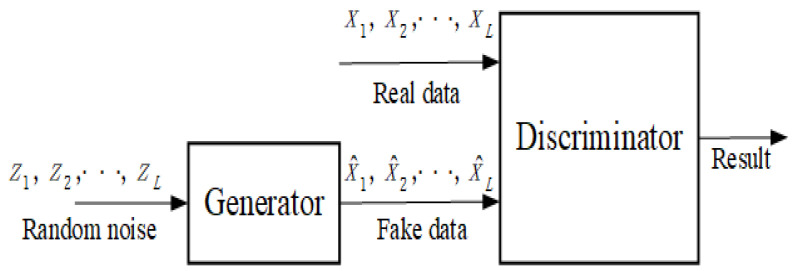
The basic structure of the GANs.

**Figure 2 entropy-25-00052-f002:**
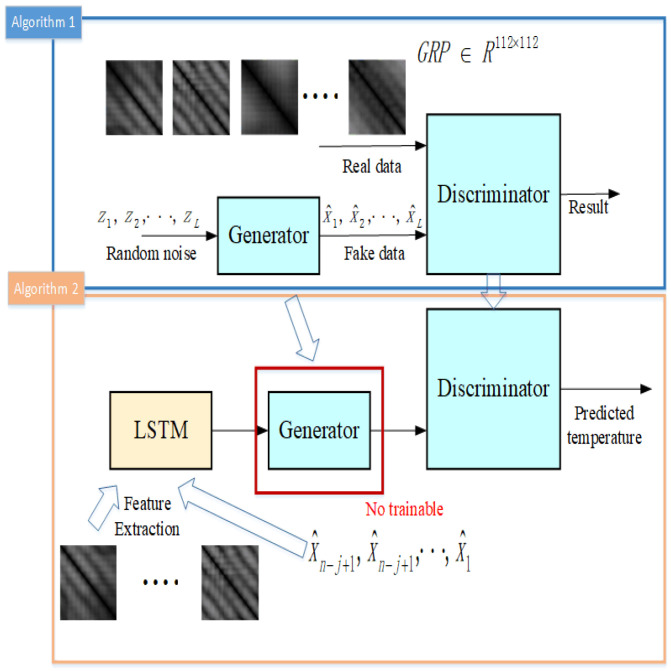
The structure of our proposed model.

**Figure 3 entropy-25-00052-f003:**
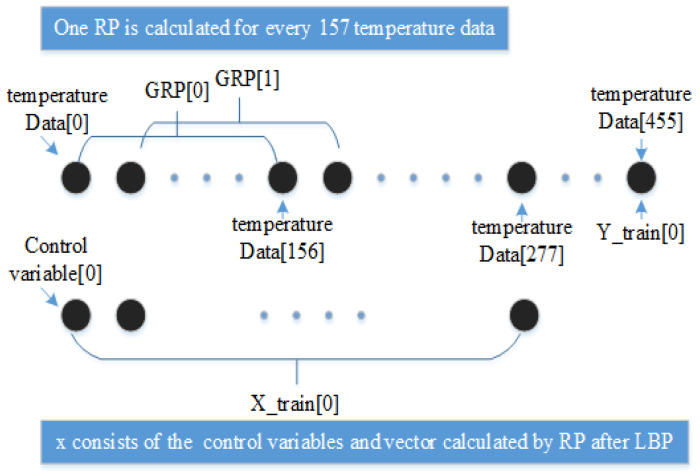
Making datasets for neural networks.

**Figure 4 entropy-25-00052-f004:**
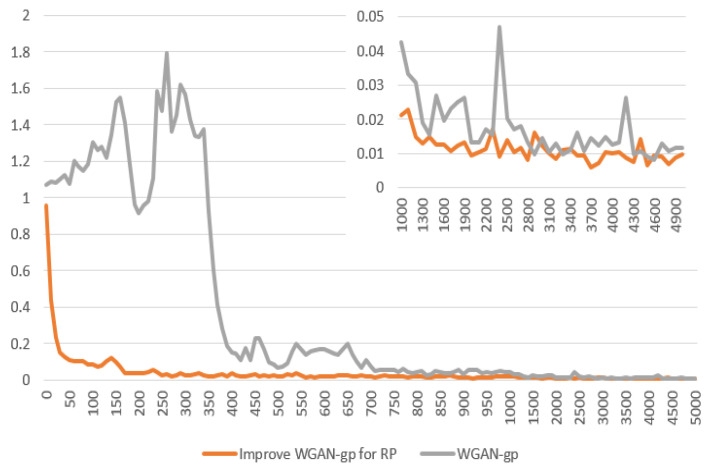
SD for WGAN-gp and Improved WGAN-gp for GRP. (*X*-axis is iteration, *Y*-axis is SD).

**Figure 5 entropy-25-00052-f005:**
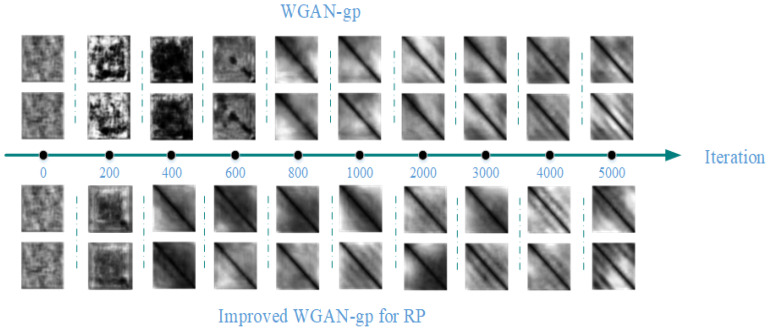
Samples generated by WGAN-gp and Improved WGAN-gp for RP. The moments for which the images are generated are the 0th, 200th, 400th, 600th, 800th, 1000th, 2000th, 3000th, 4000th, and 5000th iteration.

**Figure 6 entropy-25-00052-f006:**
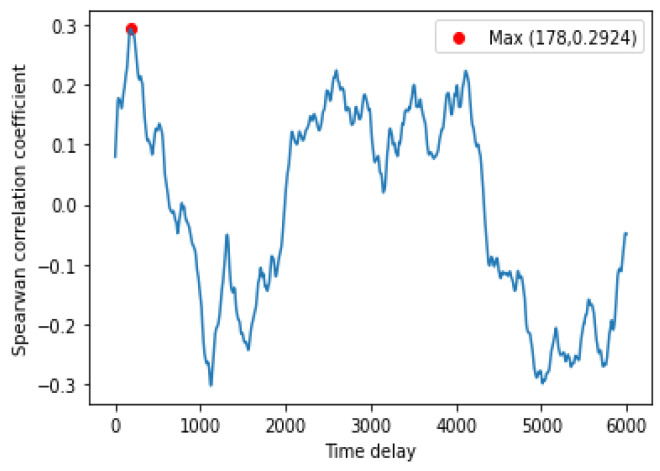
Spearman coefficients of temperature series with different time delays and levels of coal consumption (when the time delay is taken as 178, the a factor obtains the maximum value 0.2924).

**Table 1 entropy-25-00052-t001:** Descriptive statistical of the data set.

	Mean	Std	Min	Max
temperature data	610.93	42.86	479.99	739.51

**Table 2 entropy-25-00052-t002:** Comparison of prediction performances of other models and proposed model.

Method	Logcosh	Mean Square Error	Absolute Temperature Error (°C)
PSR-LSTM	$1.48\times 10^{-2}$	$3.02\times 10^{-2}$	43.45
GRP-LSTM	$1.78\times 10^{-4}$	$3.57\times 10^{-4}$	4.73
GRP-LSTM -Autoencoder	$9.83\times 10^{-5}$	$1.96\times 10^{-4}$	3.51
**Proposed** **model**	$8.03\times 10^{-5}$	$1.61\times 10^{-4}$	3.16

**Table 3 entropy-25-00052-t003:** Comparing the effect of different time delays on prediction accuracy.

**Time Delay**	188	**Our Model**	168	158	98	38
**Logcosh**	$1.94\times 10^{-4}$	$8.03\times 10^{-5}$	$1.39\times 10^{-4}$	$9.88\times 10^{-5}$	$1.54\times 10^{-4}$	$1.26\times 10^{-4}$
**Mean** **Square** **Error**	$3.87\times 10^{-4}$	$1.61\times 10^{-4}$	$2.78\times 10^{-4}$	$1.98\times 10^{-4}$	$3.07\times 10^{-4}$	$2.52\times 10^{-4}$
**Absolute** **Temperature** **Error (*C°*)**	4.92	**3.17**	4.16	3.51	4.38	3.97

**Table 4 entropy-25-00052-t004:** Characteristics of different LBP models.

Default	Original LBP, which is gray-scale but not rotation-invariant [23].
Ror	Extension of default implementation, which is rotation-invariant and gray-scale.
Uniform	Improved rotation invariance with uniform patterns and finer quantization of the angular space which, is gray-scale and rotation-invariant [23].
Nri-uniform	Non rotation-invariant uniform patterns variant which is only gray-scale invar- iant [24,25].
Var	rotation-invariant variance measures of the contrast of local image texture which is rotation but not gray scale invariant.

**Table 5 entropy-25-00052-t005:** Comparing the effects of different GRP feature-extraction methods on forecast precision.

Pattern	Logcosh	Mean Square Error	Absolute Temperature Error ($C^{\circ }$)	Feature Vector Dimension
Ror	$1.63\times 10^{-4}$	$3.25\times 10^{-4}$	4.92	256
**Default**	$8.03\times 10^{-5}$	$1.61\times 10^{-4}$	**3.17**	256
Uniform	$1.15\times 10^{-4}$	$2.31\times 10^{-4}$	4.16	10
Nri-uniform	$1.07\times 10^{-4}$	$2.14\times 10^{-4}$	3.51	59
Var	$1.84\times 10^{-4}$	$3.69\times 10^{-4}$	4.38	59

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
