# Peer review of "Forecasting for Chaotic Time Series Based on GRP-lstmGAN Model: Application to Temperature Series of Rotary Kiln"

_entropy, 2022, doi:10.3390/e25010052_

Round 1
Reviewer 1 Report
The paper is nice and written well. It can be considered for publication with some minor comments:-
1- all equations should be ended by , or .
2- most of equations should be written in one line.
3- Many types errors as e adversarial network(GAN) make space between e adversarial network and (GAN) and many errors as this
4- List of abbreviations should be added
Author Response
Response to Reviewer 1 Comments
Point 1: all equations should be ended by , or .
Response 1: Thank you for careful reading. We are sorry for the mistake, which have been corrected accordingly.
Point 2: most of equations should be written in one line.
Response 2: Thank you for careful reading. We are sorry for the unpleasant reading experience caused by the unreasonable layout, we have changed the layout of the formula.
Point 3: Many types errors as e adversarial network(GAN) make space between e adversarial network and (GAN) and many errors as this.
Response 3: Thank you for careful reading. We are sorry for the mistake, which have been corrected accordingly.
Point 4: List of abbreviations should be added.
Response 4: Thank you very much for your comments. Your suggestions are helpful for us to improve the quality of this paper. We added List of abbreviations to the third page .
Reviewer 2 Report
Overall, the present work is well done and well structured. The work is an original piece of science explaining in an innovative way a new methodology to process experimental time-series of temperature measurements by using neural network methodology enriched with image/pattern recognitions.
The results are coherent with the experimental observations and the methodology is explained in all its details. The authors were able to clearly illustrate the problem and to build an innovative algorithm that is able to evaluate experimental results with a not-negligible accuracy.
Therefore, I strongly suggest this paper for publication in the Journal Entropy as in the present form.
Author Response
Response to Reviewer 2 Comments
Response: Thank you for your recognition of my work, which is a great encouragement to me.
Reviewer 3 Report
In this paper, a GRP-lstmGAN, a Global Recurrence Plot (GRP) based Generative Adversarial Network (GAN) and Long Short-Term Memory (LSTM) combination method that can effectively display important information about time scales, is proposed.
The data is first subjected to a series of pre-processing operations such as data smoothing.
Then, GRP uses global and local information from time series to transform one-dimensional time series into two-dimensional images.
Finally, a combination of LSTM and improved GAN models was used to predict temperature time series.
First of all, I find this combination interesting. However, the main problem I see is explainability. It would be nice to have some explainable concepts, such as which part of the time series is relevant to the understudied problem, either from the recurrence plot or from the raw time series. I think it is important to emphasise these aspects.
Is it possible to get other transformations of data other than GRP?
Another important point is the code developed during this work. It is helpful for reviewers to see what you did and also for reproduction purpose.
A final minor comment, there are too many typos (even repeated words), please take the time read it again.
Author Response
Response to Reviewer 3 Comments
Point 1: First of all, I find this combination interesting. However, the main problem I see is explainability. It would be nice to have some explainable concepts, such as which part of the time series is relevant to the understudied problem, either from the recurrence plot or from the raw time series. I think it is important to emphasise these aspects.
Response 1: Thank you for careful reading. We add some explanations in introducation and Section 3. Let me explain to you in detail the advantages of our method over traditional time series prediction.
Sintering temperature is determined by many variables, and the dynamic characteristics of chaotic time series can not be fully reflected only by sintering temperature itself. Using phase space reconstruction (PSR), we can reconstruct an equivalent system in state space by using the observed chaotic time series. Time delay and dimension embedding can be used to reconstruct the influence of other variables on the system.
Rotary kiln is a system with large time delay, so the traditional time series prediction method can not consider both local and global information at the same time. Global Recurrence Plot (GRP) is based on PSR, which can be exploited to characterise the system’s behaviour in phase space. It is equivalent to solving the distance matrix of different states after PSR. All relevant dynamical information is fully preserved in the distance matrix[1].
Finally, compared with the traditional prediction method based on time series itself. Our method can extract nonlinear and dynamic features of chaotic time series.
Point 2: Is it possible to get other transformations of data other than GRP?
Response 2: Thank you for you suggestion. I also considered this problem in my research. There are many ways to convert time series into images. But because the sintering temperature of rotary kiln is chaotic time series, GRP is the most suitable analysis method for chaotic time series
Point 3: Another important point is the code developed during this work. It is helpful for reviewers to see what you did and also for reproduction purpose.
Response 3: I'm sorry that my code may not help you much. Some of the code was completed in the previous work, so the code includes C,python, Cpython and matlab. I did not sort out the annotations, so it was difficult to read them.
Point 4: A final minor comment, there are too many typos (even repeated words), please take the time read it again.
Response 4: Thank you for careful reading. We are sorry for the mistake, which have been corrected accordingly.
[1] G. McGuire, N.B. Azar, M. Shelhamer, Recurrence matrices and the preservation of dynamical properties, Phys. Lett. A 237 (1–2) (1997) 43–47.